# Why the Psychosomatic View on Myalgic Encephalomyelitis/Chronic Fatigue Syndrome Is Inconsistent with Current Evidence and Harmful to Patients

**DOI:** 10.3390/medicina60010083

**Published:** 2023-12-31

**Authors:** Manuel Thoma, Laura Froehlich, Daniel B. R. Hattesohl, Sonja Quante, Leonard A. Jason, Carmen Scheibenbogen

**Affiliations:** 1German Association for ME/CFS, 20146 Hamburg, Germany; daniel.hattesohl@dg.mecfs.de (D.B.R.H.); sonja.quante@dg.mecfs.de (S.Q.); 2Research Center CATALPA, FernUniversität in Hagen, 58097 Hagen, Germany; laura.froehlich@fernuni-hagen.de; 3Center for Community Research, DePaul University, Chicago, IL 60614, USA; ljason@depaul.edu; 4Institute of Medical Immunology, Charité—Universitätsmedizin Berlin, Corporate Member of Freie Universität Berlin and Humboldt Universität zu Berlin and Berlin Institute of Health (BIH), 10117 Berlin, Germany; carmen.scheibenbogen@charite.de

**Keywords:** myalgic encephalomyelitis, chronic fatigue syndrome, ME/CFS

## Abstract

Since 1969, Myalgic Encephalomyelitis/Chronic Fatigue Syndrome (ME/CFS) has been classified as a neurological disease in the International Classification of Diseases by the World Health Organization. Although numerous studies over time have uncovered organic abnormalities in patients with ME/CFS, and the majority of researchers to date classify the disease as organic, many physicians still believe that ME/CFS is a psychosomatic illness. In this article, we show how detrimental this belief is to the care and well-being of affected patients and, as a consequence, how important the education of physicians and the public is to stop misdiagnosis, mistreatment, and stigmatization on the grounds of incorrect psychosomatic attributions about the etiology and clinical course of ME/CFS.

## 1. Introduction

Myalgic Encephalomyelitis/Chronic Fatigue Syndrome (hereafter: ME/CFS) is a chronic and debilitating disease that predominantly affects women, but also men, in all age groups [1]. The defining symptom of ME/CFS is post-exertional malaise (PEM), a marked worsening of existing symptoms even after minor physical and/or mental exertion. PEM usually occurs immediately or up to 12–48 h after the triggering activity, which can be physical, cognitive, or emotional in nature [2]. PEM differentiates ME/CFS from other fatiguing illnesses, such as multiple sclerosis or Sjögren’s syndrome [3,4]. Another typical symptom of ME/CFS is orthostatic intolerance, a circulatory disturbance while sitting and standing, which, especially in more severe cases, can result in patients being bed-bound [5]. Further key symptoms are chronic fatigue, neurocognitive symptoms often described as brain fog, pain (head and muscle aches), and sensitivity to light and sound [6]. Before the COVID-19 pandemic, an estimated 0.2–0.5 percent of the general population was affected by ME/CFS [7,8]. As a significant subgroup of Long COVID patients meets the diagnostic criteria of ME/CFS [9,10], the prevalence of ME/CFS has likely increased substantially since the beginning of the COVID-19 pandemic [11,12]. While tests like a 2-day cardiopulmonary exercise test [13,14] or a hand-grip strength test [15,16] aid in the diagnosis, broadly validated biomarkers for ME/CFS are not established yet, and ME/CFS remains a clinical diagnosis. The Canadian Consensus Criteria [6], the International Consensus Criteria [17], and the Institute of Medicine Criteria [18] are the current and established diagnostic criteria for ME/CFS used in research and clinical care. Although a large body of scientific literature shows abnormalities in the immune system [19,20,21,22], the vascular system [5,20,23], and the energy metabolism [24,25], a still widespread misconception about ME/CFS is the idea that the disease is psychosomatic in its etiology or that psychological factors play a key role in the chronic nature of the disease. Contrary to psychosomatic hypotheses, replicable organic abnormalities are evident in ME/CFS [26]. The most important replicated abnormalities include a significant reduction in cerebral blood flow [27,28,29], endothelial dysfunction [30,31], a reduction in systemic oxygen supply [32,33], a reduced peak oxygen consumption [34], an increase in ventricular lactate levels [35], hypometabolism [36], and increased levels of autoantibodies against G-protein-coupled receptors [37,38,39]. Many organic abnormalities found in ME/CFS correlate with symptom severity, indicating a relevant role in the disease process [29,31,37,40]. Moreover, psychological factors did not predict which individuals developed ME/CFS in a prospective study [41]. Many studies indicate that in the majority of ME/CFS cases, the disease begins with a viral infection, such as glandular fever, influenza (flu), or COVID-19. Therefore, infectious diseases are considered proven disease triggers of ME/CFS [12,42]. Lack of knowledge among medical professionals about the etiology, diagnostics, and treatment of ME/CFS [43,44] and the misclassification of ME/CFS as a psychosomatic illness often accompanied by the denial of the existence of ME/CFS as a clinical entity still prevents the disease from being diagnosed and adequately treated [45]. As a result, many affected individuals have not received an ME/CFS diagnosis even years after the onset of their illness [46]. In addition, the poor recognition of ME/CFS, along with incorrect psychosomatic causal attributions, has impeded basic biomedical research and clinical studies over decades, so there is still no approved drug for ME/CFS.

## 2. The Attribution of Organic Diseases to Psychosomatic Factors Has a Long Tradition

While various definitions of the term “psychosomatic illness” exist, a common denominator is the idea that they are physical syndromes without relevant organic correlates and are thus mainly, or in a relevant part, caused and perpetuated by psychosocial factors [47,48,49]. Attributing organic diseases to psychological causes has actually been documented at virtually every point in medical history [50]. Several, if not all, organic diseases known today have, at some point in history, been attributed to psychosomatic factors. As an illustrating example, Franz Alexander, one of the founders of modern psychosomatic medicine, postulated a list of seven illnesses (later described as the “holy seven”) that he characterized as psychosomatic [51]. The list consists of diseases such as rheumatoid arthritis, asthma, and hyperthyroidism. Today, as scientific findings have uncovered somatic causes of those diseases, claims that these are psychosomatic are no longer accepted by the medical community. For diseases that are less well-studied or whose research findings are less well known, theories of psychosomatic etiology remain popular. In addition to ME/CFS, this is also the case, for example, for fibromyalgia [52], irritable bowel syndrome [53], or endometriosis [54].

## 3. Psychosomatic Disease Models for ME/CFS Are in Contradiction with Scientific Findings, Expert Opinions, and the Lived Experiences of Patients

The extent and nature of the attribution of ME/CFS to psychosomatic factors have changed within the last 50 years [55]. In a publication in the British Medical Journal, McEvedy and Beard [56] viewed ME/CFS as epidemic hysteria on the grounds that—among other reasons—ME/CFS predominantly affected women (including nurses). However, a recent reanalysis of the data showed that the occurrence of ME/CFS at that time followed the usual spread of infectious diseases [57]. Due to the sexist connotation (hysteria is derived from the ancient Greek word hystéra, translated to uterus) and the unclear definition, the term hysteria is no longer used today and is also no longer part of the International Classification of Diseases.

More recent psychosomatic theories, such as those put forward by the authors of the PACE study published in 2011, focus on so-called “dysfunctional cognitions and behaviors” that are assumed to maintain the symptoms. Specifically, in the theoretical assumptions of the PACE study, fear of activity, resulting activity-avoiding behaviors, and deconditioning are held responsible for the maintenance of ME/CFS symptoms [58]. Other psychosomatic disease models [49] use additional factors, such as false illness convictions, symptom expectations causing the sensation of the symptom to manifest, and alleged operant conditioning on being sick through health care experiences (“secondary gain”). Today, scientific papers portraying ME/CFS as psychosomatic continue to be published [59]. Many physicians still believe that ME/CFS is a psychosomatic disease [60], and 90% of patients with ME/CFS are at least once told by health professionals that their symptoms are psychosomatic before receiving an ME/CFS diagnosis [61]. In stark contrast to these widespread beliefs, empirical evidence does not support a psychosomatic etiology of ME/CFS [41]. There is also a broad consensus among patients with ME/CFS, as well as ME/CFS experts, that psychosomatic factors do not play a relevant role in the etiology as well as the clinical course of ME/CFS [62,63]. Psychosomatic etiology, such as the deconditioning hypothesis presented in the cognitive-behavioral model of ME/CFS [64], is assumed to involve cognitive appraisals that create an inaccurate perception of being sick and fatigued without a persistent physical cause. Consequently, the model assumes that patients with ME/CFS reduce their physical activity level, which in turn worsens their functional status [65]. Research conducted in this tradition has thus recommended that in order to get better, patients with ME/CFS should change their dysfunctional cognitions with Cognitive–Behavioral Therapy (CBT) and should gradually increase their activity level with Graded Exercise Therapy (GET) [64,65]. It has been shown that the tenets of the cognitive–behavioral model of ME/CFS are not in line with evidence of physical abnormalities in ME/CFS [66,67,68], especially concerning the findings of an abnormal reaction to even minor exertion [34]. Therefore, the idea that patients with ME/CFS can improve their symptoms by changing their “dysfunctional cognitions”, as well as by increasing their activity level, conveys a harmful message; namely, that if symptoms do not improve or even worsen with increased activity, it is somehow the responsibility of the patients who did not work hard enough to overcome their “dysfunctional cognitions” [67]. One major problem of this psychosomatic view of ME/CFS is that it is contradictory to pacing as the central disease management strategy for ME/CFS. Pacing enables patients to allocate their limited available energy in such a way that, as far as possible, no worsening of symptoms occurs due to the leading symptom of ME/CFS, PEM [69,70,71]. In addition to planning, scheduling, and dividing the activities that are still possible, pacing can also include minimizing orthostatic load by lying down [72]. In contrast, proponents of the psychosomatic approach view pacing as “activity-avoiding behaviors” that get in the way of recovery and instead favor activity-enhancing therapies such as Graded Exercise Therapy (GET) [58]. However, patient surveys show that activity-enhancing therapies like GET worsen the disease state in a large proportion of patients with ME/CFS [73]. Results of randomized controlled trials such as the PACE study suggest only minor improvements in ME/CFS through activity-enhancing therapies and have been rated as low or very low in methodological quality by the UK National Institute for Health and Care Excellence [74]. Moreover, they are in contradiction with evidence on the pathophysiology of PEM. Various studies with cardiopulmonary exercise testing repeated after 24 h (2-day CPET) have shown that PEM can be objectified through different measurements, validating patients’ experience of this disabling symptom [75]. Stussmann et al. [2] have thoroughly analyzed how patients experience PEM. During a 2-day CPET, in contrast to healthy controls, the anaerobic threshold and maximal oxygen capacity of patients with ME/CFS dropped significantly at the second exercise test [76]. Thus, patients with ME/CFS have significant difficulties with aerobic energy production during repeated (over)exercise and more quickly switch to the significantly more inefficient lactate-producing anaerobic energy production. Moore et al. [77] found that patients with ME/CFS need, on average, two weeks to recover from a 2-day CPET compared to only two days for sedentary controls. Further studies with different exercise tests found changes in gene expression post-exercise in patients with ME/CFS [78,79] and a lack of reduction of arterial stiffness, possibly indicating a lack of vasodilation during and after exercise in patients with ME/CFS [80]. Ghali et al. [81] found that elevated lactate levels in a resting state correlate with more severe PEM episodes. Together, these findings also contradict the hypothesis that ME/CFS is caused by deconditioning. For instance, van Campen et al. [82] showed that decreased cerebral blood flow in ME/CFS in the upright position is independent of physical fitness and thus independent of deconditioning. Therefore, central physiological abnormalities in ME/CFS cannot be attributed to deconditioning but can be explained by pathological disease processes. In sum, the evidence does not align with the central claim of psychosomatic disease models of ME/CFS, which assumes that dysfunctional cognitions and the resulting deconditioning explain the etiology and maintenance of ME/CFS symptoms [67,68].

It has been shown that the cognitive–behavioral model of ME/CFS contributes to the stigmatization of patients with ME/CFS due to assigning them responsibility for the persistence of their symptoms, which in turn can aggravate their impaired functional status and can be a burden for social relationships. Qualitative and quantitative evidence show that psychosomatic attributions by physicians, family members, or acquaintances correspond with higher perceived stigma and lower life satisfaction and relationship satisfaction in patients [44,83,84,85,86,87]. Moreover, psychosomatic attributions by physicians can result in medical gaslighting, where physicians tell the patients that they are not seriously ill and are making the symptoms up [45,88,89]. Psychosomatic misattribution can also result in the prescription of harmful therapies such as GET, which can induce potentially permanent worsening of symptoms due to PEM [89,90]. The attribution of ME/CFS symptoms to psychosomatic causes is not the only form of stigmatization and mistreatment patients with ME/CFS experience [89], but it is still deeply entrenched institutionally through the discipline of psychosomatic medicine and thus very influential [91]. Students are presented with psychosomatic disease models for unexplained syndromes already in medical school, and only a minority of medical schools include ME/CFS in their curricula [92]. The negative and skeptical attitude of many physicians towards ME/CFS is also reinforced by the fact that some physicians without clinical experience with ME/CFS spread psychosomatic theories in media appearances or scientific articles. Recently, this also became particularly evident in the public and scientific discourse on Long COVID [93,94].

## 4. The Vast Majority of Research Views ME/CFS as an Organic Disease

A systematic review by Muller et al. [95] shows that a clear majority of primary studies on causal factors of ME/CFS investigate organic factors, and only around 20% investigate psychological factors (including studies published between 1979 and 2019). Accordingly, a content analysis by Siegel et al. [96] of 241 American newspaper articles published between 1987 and 2013 shows that 65% portray the etiology of ME/CFS as organic, 22% as both organic and psychogenic, and only 3% as strictly psychogenic. In line with this research, clinicians specialized in ME/CFS usually see ME/CFS as an organic disease (e.g., the U.S. ME/CFS Clinician Coalition [62]). The same is true for the vast majority of patients and their organizations [63]. Nevertheless, a vocal minority of researchers remains convinced of a psychosomatic (co-)causation of ME/CFS despite the frequently demonstrated organic abnormalities and the simultaneous lack of evidence for relevant psychosomatic factors. The striking discrepancy between the strong conviction among proponents of a psychosomatic etiology of ME/CFS and the simultaneous lack of evidence for this view has also been observed in other scientific fields. Research shows that across different topics such as homeopathy, vaccination, or COVID-19, individuals who strongly disagree with the scientific consensus are, on average, less knowledgeable about the topics than others but are more convinced of their knowledge [97]. Together with the known lack of knowledge about ME/CFS by physicians [45,60], this makes it necessary for political and medical institutions to broadly inform physicians and the public about the disease in order to counter misinformation and prevent patients from being stigmatized, misdiagnosed, and mistreated based on incorrect psychosomatic theories. Expert-led webinars have been shown to be a feasible approach to educating physicians about the nature of ME/CFS, resulting in physicians making fewer incorrect psychosomatic attributions of ME/CFS in a knowledge test after attending the webinar [98].

## 5. Chronic Diseases Include the Risk of Mental Comorbidities

Similar to other chronic diseases that can decrease patients’ quality of life, the psychological burden of ME/CFS is enormous, and secondary mental illnesses such as reactive depression may occur. However, as Jason et al. [41] show in a prospective study, patients with ME/CFS are no more likely to have a history of mental illness before the onset of their disease than the average population. Since some of the symptoms of depression and ME/CFS overlap (e.g., fatigue), ME/CFS patients can be misdiagnosed with depression [99]. ME/CFS and depression can be differentiated based on symptomatology [100]. Important symptoms to differentiate the illnesses include, first and foremost, PEM: exercise worsens the symptoms for several days to weeks in patients with ME/CFS [77] but not in patients with major depression. While a portion of the organic abnormalities seen in ME/CFS also occurs in depression (e.g., endothelial dysfunction) [101], other indicators like heartrate variability profiles can differentiate between the illnesses [102]. In addition to ME/CFS itself, the lack of medical care and social support is particularly burdensome for those affected [43,103]. In the health care system, patients with ME/CFS have to reckon with medical gaslighting and sometimes severe maltreatment—physicians convince patients that they are imagining the disease, that they are overdramatizing the situation, or that they are psychosomatically ill [44]. When admitted to hospital or rehabilitation programs, patients with ME/CFS who are wrongly classified as having a psychosomatic illness are threatened with mistreatment, including activity-increasing therapies like GET that can seriously harm them [8]. Impressive reports of such incidents can be found, for example, in newspaper articles [104,105] or in a report by #MEAction [106].

The extremely precarious situation in which patients with ME/CFS are often left to their own devices can make supportive psychotherapy useful if desired by patients, and if no lasting deterioration of the condition due to physical or mental overexertion resulting in PEM is to be expected. This can only be achieved if the treating therapist is familiar with ME/CFS. Grande et al. [107] describe how supportive psychotherapy in ME/CFS should look like. It is essential that ME/CFS is recognized as an organic disease and that those affected are supported in the implementation of pacing as a disease management strategy. In such a setting of psychotherapy, patients can be helped to find ways to cope with the immense suffering and limitations caused by ME/CFS.

## 6. How to Best Help Affected Patients

Due to the lack of knowledge about ME/CFS among health care professionals and its frequent and incorrect classification as a psychosomatic illness, it is extremely difficult for affected persons to receive not only appropriate diagnostics and treatment but also nursing care, disability benefits, pensions, or social benefits. To improve this situation in the future and to provide adequate research funding and medical care for ME/CFS, it is important to educate physicians and the public about how wrong and harmful false psychosomatic disease models are in ME/CFS. It is, therefore, encouraging that the problem of incorrect psychosomatic attribution of ME/CFS seems to have reached the political arena. The COVID-19 Special Report of the European Parliament, adopted in July 2023 [108], states that patients with ME/CFS are often psychosomatically misdiagnosed, which, in addition to stigmatization, can result in harmful mistreatment. Additionally, more and more representatives of psychology, psychiatry, and psychosomatic medicine have started to recognize and address this problem. The integration of supportive psychotherapy on the grounds of a biomedical understanding of ME/CFS into interdisciplinary teams would represent an important step forward in terms of comprehensive care for ME/CFS patients [107]. However, there is still a lot of educational work to be done before this concept reaches the daily care of patients with ME/CFS.

## 7. Conclusion: False Psychosomatic Attributions on the Etiology of ME/CFS Hinder Adequate Patient Care

Even though, in recent years, more and more healthcare practitioners view ME/CFS correctly as a somatic illness, there are still widespread views in the medical community of ME/CFS being a psychosomatic illness. These views are detrimental for affected patients as they can result in misdiagnosis and harmful therapies, such as GET, which can result in lasting worsening of symptoms. Moreover, false psychosomatic attributions lead to stigmatization. A correct biomedical understanding of ME/CFS in line with the current state of empirical evidence for treatment is, therefore, essential to providing adequate care to patients with ME/CFS.

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
