# Peer review of "Why the Psychosomatic View on Myalgic Encephalomyelitis/Chronic Fatigue Syndrome Is Inconsistent with Current Evidence and Harmful to Patients"

_medicina, 2023, doi:10.3390/medicina60010083_

Round 1
Reviewer 1 Report
Comments and Suggestions for Authors
The Review deals with the problem of incorrect psychosomatic attribution of ME/CFS. The article is well-written, but should definitely be expanded, as the topic of the great importance, and some critical aspects should be discussed in more details.
1. Line 49 Please, specify the most important replicable organic abnormalities in ME/CFS (not in general terms, such as “abnormalities in the immune system, the vascular system, and the energy metabolism”).
2. Please, provide evidence that ME/CFS is not a psychosomatic illness. It is known, that organic abnormalities, which have been described in ME/CFS - such as neuroinflammation (Gritti D, Delvecchio G, Ferro A, Bressi C, Brambilla P. Neuroinflammation in Major Depressive Disorder: A Review of PET Imaging Studies Examining the 18-kDa Translocator Protein. J Affect Disord. 2021 Sep 1;292:642-651. doi: 10.1016/j.jad.2021.06.001), chronic production of pro-inflammatory cytokines (Wang, H., Li, P., Zhang, Y., Zhang, C., Kangwei, L., & Song, C. (2020). Cytokine changes in different types of depression: Specific or general? Neurology Psychiatry and Brain Research, 36, 39-51.), HPA-axis abnormalities (Herane-Vives A, Papadopoulos A, de Angel V, Chua KC, Soto L, Chalder T, Young AH, Cleare AJ. Cortisol levels in chronic fatigue syndrome and atypical depression measured using hair and saliva specimens. J Affect Disord. 2020 Apr 15;267:307-314. doi: 10.1016/j.jad.2020.01.146.), cardiovascular abnormalities (Shinba T, Kuratsune D, Shinba S, Shinba Y, Sun G, Matsui T, Kuratsune H. Major Depressive Disorder and Chronic Fatigue Syndrome Show Characteristic Heart Rate Variability Profiles Reflecting Autonomic Dysregulations: Differentiation by Linear Discriminant Analysis. Sensors. 2023; 23(11):5330. https://doi.org/10.3390/s23115330) are typical also for mental disorders. So, the existence of these abnormalities in ME/CFS are not enough to conclude that ME/CFS is not a mental disorder.
If there is not enough evidence to prove that ME/CFS is not a mental disorder, maybe, in light of the above, it would be more consistent to claim that all mental disorders are not psychosomatic, but organic.In this case the authors should focus on the vulnerability of the the concept of “psychosomatic disorders” itself.
3. Line 134 - please, provide reference “For instance, van Campen et al. showed that decreased cerebral blood 134 flow in ME/CFS in the upright position is independent of physical fitness.” It should be [5].
4. Psychosomatic concept is not very popular now. Instead of that, many physicians are convinced that patients who believe they have ME/CFS, actually suffer from a depression disorder or a somatic symptom disorder. As Prof. Jason is a prominent specialist on the differentiation of ME/CFS from different mental disorders, it will be a great assistance if he expands on these issues from the practical, clinical point of view.
Reviewer 2 Report
Comments and Suggestions for Authors
This paper intends to present the argument that ME/CFS can no longer be viewed as psychosomatic in origin as it ignores key features like PEM, cognitive symptoms and orthostatic intolerance. Assumptions for management if it were psychosomatic do nothing for these symptoms and may make them worse.
The paper describes the evidence for an organic basis and inconclusive research based on a psychosomatic model but I am not sure that the paper demonstrates clearly the harm to patients although that can be implied. A clearer outline of harm such as ignoring PEM and its impact, neglecting orthostatic symptoms, attributing unfounded assumptions about patient motivation or attitudes and ignorance of the features of ME/CFS hence delay in diagnosis as well as prominent voices advocating psychosomatic diagnosis would be helpful. These issues are raised in para 3 but resulting harm is implied without a lot of evidence. The current conclusion should be part of the argument as new information is presented. A new conclusion that is more succinct is needed.
Hence a section demonstrating the overt harm to patients more clearly or modifying the title should occur. For example, 'ME/CFS viewed as a psychosomatic illness ignores the evidence and can harm the patient'.
There are a couple of typographical errors (line 319, 343)
Comments on the Quality of English LanguageQuality of English language is good.
Round 2
Reviewer 1 Report
Comments and Suggestions for Authors
It is an extremely valuable manuscript which deals with an important medical and social problem. Highly recommended for publication